# Investigation of Potential Drug Targets for Cholesterol Regulation to Treat Alzheimer’s Disease

**DOI:** 10.3390/ijerph20136217

**Published:** 2023-06-24

**Authors:** Marina Passero, Tianhua Zhai, Zuyi Huang

**Affiliations:** Department of Chemical Engineering, Villanova University, Villanova, PA 19085, USA

**Keywords:** Alzheimer’s disease, cholesterol biosynthesis, protein-protein interaction, amyloid beta, protein target, ligand-protein docking, drug discovery

## Abstract

Despite extensive research and seven approved drugs, the complex interplay of genes, proteins, and pathways in Alzheimer’s disease remains a challenge. This implies the intricacies of the mechanism for Alzheimer’s disease, which involves the interaction of hundreds of genes, proteins, and pathways. While the major hallmarks of Alzheimer’s disease are the accumulation of amyloid plaques and tau protein tangles, excessive accumulation of cholesterol is reportedly correlated with Alzheimer’s disease patients. In this work, protein-protein interaction analysis was conducted based upon the genes from a clinical database to identify the top protein targets with most data-indicated involvement in Alzheimer’s disease, which include ABCA1, CYP46A1, BACE1, TREM2, GSK3B, and SREBP2. The reactions and pathways associated with these genes were thoroughly studied for their roles in regulating brain cholesterol biosynthesis, amyloid beta accumulation, and tau protein tangle formation. Existing clinical trials for each protein target were also investigated. The research indicated that the inhibition of SREBP2, BACE1, or GSK3B is beneficial to reduce cholesterol and amyloid beta accumulation, while the activation of ABCA1, CYP46A1, or TREM2 has similar effects. In this study, Sterol Regulatory Element-Binding Protein 2 (SREBP2) emerged as the primary protein target. SREBP2 serves a pivotal role in maintaining cholesterol balance, acting as a transcription factor that controls the expression of several enzymes pivotal for cholesterol biosynthesis. Novel studies suggest that SREBP2 performs a multifaceted role in Alzheimer’s disease. The hyperactivity of SREBP2 may lead to heightened cholesterol biosynthesis, which suggested association with the pathogenesis of Alzheimer’s disease. Lowering SREBP2 levels in an Alzheimer’s disease mouse model results in reduced production of amyloid-beta, a major contributor to Alzheimer’s disease progression. Moreover, its thoroughly analyzed crystal structure allows for computer-aided screening of potential inhibitors; SREBP2 is thus selected as a prospective drug target. While more protein targets can be added onto the list in the future, this work provides an overview of key proteins involved in the regulation of brain cholesterol biosynthesis that may be further investigated for Alzheimer’s disease intervention.

## 1. Introduction

Alzheimer’s disease is a progressive neurodegenerative disease caused by the accumulation of amyloid-beta plaques and tau protein tangles, which lead to neuronal and synaptic loss [1,2]. It is the most common cause of dementia and is difficult to diagnose due to the fact that many of the symptoms are similar to the normal effects of aging [3]. Research on Alzheimer’s disease has increased over the years since this disease worsens over time and has no cure [4]. Excessive cholesterol has been found to accumulate in the brain of those with Alzheimer’s disease, specifically in the myelin, astrocytes, and neurons [5]. Cholesterol can be synthesized in both the blood and the brain, but the majority of biosynthesis in the adult brain takes place in astrocytes and is then transported to the neurons via ApoE [6,7,8]. The blood brain barrier separates cholesterol metabolism in the brain from the rest of the body [9]. Brain cholesterol synthesis is similar to that of blood cholesterol synthesis, starting with de novo synthesis in which acetyl-CoA is converted into HMG-CoA [10]. The rate-limiting enzyme HMGCR catalyzes the reduction of HMG-CoA into mevalonate [11]. A few more reactions take place, leading to the production of zymosterol, which proceeds either through the Bloch pathway or Kandustch-Russel pathway; both pathways result in the production of cholesterol. Various proteins contribute to the metabolism of cholesterol in the brain, which offer potential targets for Alzheimer’s treatment. In the astrocytes, SCAP binds to SREBP2 in order to transport the complex from the endoplasmic reticulum to the Golgi apparatus [12]. Cholesterol is secreted out of the astrocytes by the ABCA1 transporter [13]. Lipoprotein receptors such as TREM2 internalize cholesterol into the microglia, while lipoprotein receptors such as LRP1 internalize cholesterol into the neuron [14,15]. The gene CYP46A1 converts excess cholesterol to 24S-OHC [16]. While cholesterol in the blood is dependent on the balance between dietary and hepatic synthesis, cholesterol in the brain is mainly synthesized in situ by astrocytes and oligodendrocytes. As a person ages, his/her brain cholesterol synthesis rate declines and the blood brain barrier loses its integrity [17]. Studies have also found that cholesterol has a direct effect on the production of amyloid-beta and tau proteins [18,19,20]. The addition of cholesterol to the brain promotes the activity of BACE1, secretase, and APP, which all aid in amyloid-beta production. The reduction of cholesterol esters in the brain has been found to reduce phosphorylated tau levels in mice. 

Many drugs have been developed to combat Alzheimer’s disease, but these drugs slow the progression of neurological symptoms rather than modifying the disease itself [21]. For example, Donepezil, Rivastigmine, Galantamine, Memantine, and Suvorexant are all FDA approved drugs that treat the cognitive symptoms of Alzheimer’s disease without changing the progression of the disease [22]. On the other hand, many drugs that were in clinical trials have been terminated due to their lack of improvement in cognitive function, despite their abilities to decrease amyloid-beta and phosphorylated tau accumulation [23]. Monoclonal antibodies have been developed to help the body clear amyloid-beta plaques from the brain. In 2021, the FDA approved Aduhelm for the treatment of those with early Alzheimer’s disease. This drug was effective at removing amyloid-beta plaques but was ineffective at slowing cognitive decline [24]. Lecanemab is the second monoclonal antibody approved by FDA to treat Alzheimer’s disease [25]. Concerns on side effects were raised for this drug. The only other FDA approved drugs to treat Alzheimer’s disease are Aricept, Leqemni, Namenda, Namzaric, Reminyl, and Exelon Patch. Various other monoclonal antibody drugs, as well as tau aggregation inhibitors, are currently in clinical trials. 

A number of drugs have been developed in order to inhibit target genes that play critical roles in Alzheimer’s disease pathology. Two potential target genes, GSK3B and BACE1, have been deemed possible candidates as disease-modifying agents for Alzheimer’s disease due to their involvement in amyloid-beta and tau production, as well as their ability to enhance cognitive functions [26]. In addition to these two genes, the proteins SREBP2, TREM2, LRP1, and the CYP46A1 gene are all additional candidates as targets to combat Alzheimer’s disease due to their involvement in the metabolic pathway of cholesterol in the brain [27,28,29,30].

The limited number of FDA-approved drugs to treat Alzheimer’s disease indicate the urgent need to accelerate the drug discovery pace. Since extensive evidence demonstrates the strong correlation between brain cholesterol regulation and Alzheimer’s disease [6], a detailed literature review on genes or pathways related to the brain cholesterol synthesis and regulation was conducted in this work. An interaction network was then built for those genes so that the top gene targets were identified. The involvement of these genes in Alzheimer’s disease progression was discussed in detail, which was followed by the investigation of existing clinical trials for those targets. The top drug targets were finally recommended with the consideration of existing crystal structures and the feasibility of computational screening for drug discovery. The results from this work pave the way for further drug discovery to regulate brain cholesterol to combat Alzheimer’s disease.

## 2. Materials and Methods: Identify Top Gene Targets for Cholesterol Regulation Involved in Alzheimer’s Disease

As previously mentioned in Section 1, the synthesis of cholesterol mainly takes place in the astrocytes and is further transported between the neurons and microglia. An overview of cholesterol transport in the brain is provided in Figure 1, below.

The top target genes associated with cholesterol-associated Alzheimer’s disease were determined using DisGeNET [31,32,33], the largest, most comprehensive, and free repositories of human gene-disease associations. By searching for the genes associated with both Alzheimer’s disease and HDL cholesterol measurement, the top genes were chosen based on their GDA score. The GDA score, or human gene-disease association, is a numerical value ranging from 0 to 1 that is curated from multiple databases and resources such as UniProt, Orphanet, and more. The GDA score ranks the gene-disease association according to their level of evidence based on the number and type of sources and publications supporting the association. For example, if the gene-disease association was supported by more than one curated source from either HPO, Clinvar, GWAscat, or GWAsdb, 0.1 would get added to the GDA. If the gene-disease association was supported by more than one curated source from RGD, MGD, or CTD, 0.2 would be added to the GDA. This computational trend continues for each curated source depending on how many publications support the association. Heavy research using scholarly articles from Google Scholar was performed on the top target genes associated with both Alzheimer’s disease and cholesterol metabolism in the brain. After a list of suitable and well-researched target genes were selected, the RCSB Protein Data Bank was used to confirm that a crystal structure existed for each protein. Finally, STRING [34,35,36], a free database of known and predicted protein-protein interactions that is commonly used in the Systems Biology community, was used to determine the network between the top genes and their interactions with one another. The final set of genes were chosen due to STRING’s analysis that the proteins have more interactions than expected for a random set and therefore are at least partially biologically connected as a group. The chosen target genes are ABCA1, CYP46A1, BACE1, TREM2, GSK3B, and SREBP2. In particular, ABCA1 is heavily involved in cholesterol transport and is even involved in certain pathways that regulate cholesterol synthesis. CYP46A1 controls cholesterol efflux and contributes to amyloid-beta production. BACE1 communicates with ABCA1 to cleave APP and produce amyloid plaques. TREM2 promotes cholesterol efflux and is involved in two major pathways that have been targeted for Alzheimer’s disease treatment. GSK3B is found to affect cognitive function and amyloid production, while also controlling cholesterol synthesis by activating SREBP2, which is of critical involvement in cholesterol synthesis and association with hypercholesterolemia and impaired cognition according to DisGeNET. The genes and their PDB codes along with the characteristics of their crystal structures are provided in Table 1, and the network of interactions between the six target genes are illustrated below, in Figure 2.

## 3. Results

In order to rank the six genes provided in Figure 2 as targets for intervening in Alzheimer’s disease, the reactions or pathways in which each of the six genes is involved are described in each of the following subsections in detail. Existing preclinical or clinical trials for each selected gene target are also listed for a comparison conducted in Section 4.

### 3.1. The Role of GSK3B Gene in Cholesterol Regulation Involved in Alzheimer’s Disease

GSK3B protein is a growth-signaling kinase regulated by inhibitory phosphorylation downstream of the Wnt pathway [37]. Studies have attributed this protein to having a role in memory and has been connected to Alzheimer’s disease due to the fact that its activation can lead to tau phosphorylation and amyloid-beta plaques. Two main pathways of GSK3B activation exist, including the Wnt/B-catenin pathway and the PI3K/Akt/mTor pathway (as shown in Figure 3). Through the Akt pathway, the GSK3B protein inhibits the expression of CREB, which is a binding protein that aids in cell survival and other vital functions [38]. GSK3B has also been found to phosphorylate various components of this pathway including Akt, RICTOR, TSC1 and 2, PTEN1 and 2, and others [39]. The Akt pathway is initiated by the activation of PI3K by cell surface receptors including RTK, GPCR, and cytokine receptors. Next, PIP_3_-binding proteins stimulate the synthesis of PIP_3_ and activate downstream signaling to promote cell growth, proliferation, and metabolism. Akt then phosphorylates and inhibits GSK3B, hindering its kinase activity and promoting cell survival. However, GSK3B directly phosphorylates several components including Akt. On the other hand, GSK3B simultaneously phosphorylates and destroys B-catenin throughout the Wnt pathway [39]. B-catenin plays a large role in regulating cell differentiation, proliferation, and apoptosis, and therefore has been found to enhance cognitive and memory abilities [40]. Although this pathway is not completely understood, it is known that GSK3B is associated with Axin and other molecules to create a complex that allows GSK3B to phosphorylate Axin, APC, and B-catenin. GSK3B then phosphorylates B-catenin on Thr41, Ser37, and Ser33, which allows B-TrCP to recognize B-catenin and consequently become degraded [41]. It has also been found that GSK3B can form a complex with B-catenin, therefore lowering the levels of B-catenin/TCF transcription and inhibiting Wnt signaling. Wnt signaling plays an important role in synaptic plasticity and maintenance in the adult brain. The compromised synaptic signaling is associated with Alzheimer’s disease [42].

Due to GSK3B’s role in two separate pathways involving cell proliferation, as well as its direct effect on tau phosphorylation [43] and amyloid-beta production [44], developments have been made to target GSK3B for the treatment of Alzheimer’s disease. Since the overactivity of GSK3B increases tau phosphorylation, researchers started to develop drugs that inhibit GSK3B and therefore prevent tau tangles from occurring [43]. A full list of the GSK3B inhibitors that are in preclinical or clinical trials is provided below, in Table 2. Since this work does not aim to illustrate the mechanisms of each of these inhibitors, the inhibitors of strong clinical interest are briefly illustrated here. The most successful and advanced drug thus far is Tideglusib, which is small-molecule inhibitor of GSK3B that uses the compound thiadiazolidinone to decrease amyloid deposition and lower tau phosphorylation levels [43]. In preclinical trials, this drug was found to reduce tau phosphorylation, amyloid deposition, neuron loss, and gliosis in mice [44]. Tideglusib was discontinued for Alzheimer’s treatment in 2012 after Phase 2b due to the fact that the trial missed its primary endpoint and some secondary endpoints [44]. Another GSK3B inhibitor, AL001, was developed to deliver lithium carbonate while avoiding the harmful effects of lithium [45]. Lithium inhibits GSK3B by being a direct competitor of magnesium and by decreasing phosphorylation levels through the activation of the Akt pathway [46]. AL001 is currently in Phase 3 of clinical trials [47]. Another GSK3B inhibitor that has been developed for Alzheimer’s treatment is SAR502250. This drug was found to prevent the increase in neuronal cell death and improve cognitive deficit in mice [48].

### 3.2. The Role of BACE1 Gene in Cholesterol Regulation Involved in Alzheimer’s Disease

BACE1 is another potential target gene that catalyzes the initial cleavage of APP, which is one of the earliest pathologic events in Alzheimer’s development [50]. BACE1 has been attributed as the sole initiator of amyloid-beta production, and its activity is elevated in the brains of Alzheimer’s patients [51]. It also plays a critical role in synaptic development and plasticity through multiple mechanisms such as APP cleavage, neuregulin-1 (Nrg1) cleavage, Sez6 cleavage, and Jagged-1 cleavage [52]. Nrg1 regulates myelination, the migration of glutamatergic and GABAergic neurons, and synaptic plasticity. The function of Sez6 is not fully known, but it is believed that they may act as receptors at the cell surface and exhibit adhesive and receptor trafficking functions [52]. Jagged-1 is a BACE1 substrate that regulates astrogenesis and neurogenesis through the Notch signaling pathway, as illustrated in Figure 4A. BACE1 is involved in one pathway that processes APP via the amyloidogenic pathway that involves the sequential cleavage of APP by beta- and gamma-secretase, which promotes amyloid-beta production. In this pathway, BACE1 cleaves APP in the lipid raft region to release the beta-stubs while also releasing the soluble N-terminus of APP [53]. Then CTF-beta is cleaved by gamma-secretase to release amyloid-beta into the extracellular space and the cytoplasm. The amyloid-beta released in this pathway can form amyloid plaques. The nonamyloidogenic pathway begins with the cleavage of APP by either alpha-secretase or y-secretase to release two variants of APP, sAPP-α, and sAPP-β, from the endosome or cell membrane [53]. The C90 fragment is then cleaved by γ-secretase, which releases AICD and p3 fragments [54]. BACE1 also interacts with GSK3B through the NFκB pathway. When this pathway is upregulated, GSK3B induces BACE1 expression by NFκB/p65 nuclear translocation and BACE1 promoter site binding, which increases BACE1 levels [55] (as illustrated in Figure 4C). Due to BACE1′s critical role as the initiating enzyme in amyloid-beta production, it has been selected as a prime target for slowing the progression of Alzheimer’s disease by lowering amyloid-beta levels in the brain [56].

BACE1’s involvement in the amyloidogenic pathway offers the potential to target Alzheimer’s disease. The activity of BACE1 leads to the production of amyloid-beta, so BACE1 inhibitors have been developed to prevent the accumulation of amyloid-beta plaques in the brain of Alzheimer’s patients. BACE1 inhibitors are typically small in size so that they can cross the blood brain barrier easily, which can help reduce the production of amyloid-beta in neurons [57]. One small-molecule BACE1 inhibitor that has been developed is Verubecestat, which has cellular permeability and good solubility. Preclinical tests of Verubecestat in animals demonstrated no adverse effects. Phase I trials proved that the drug was safe while effective at reducing the amyloid-beta concentration in the cerebrospinal fluid. During Phases II and III trials, participants with mild cognitive deficits were used to determine the effect Verubecestat has on cognitive and functional abilities. While the amyloid-beta plaques in their brains slightly decreased, adverse effects were reported and no improvement in cognitive function was detected. Although trial was terminated in 2018, this drug did prove that BACE1 inhibitors need to be administered several years before symptoms of Alzheimer’s disease are apparent. 

Another small-molecule BACE1 inhibitor is Lanabecestat, the inhibition potency of which is as follows: Ki = 0.4 nM for hBACE1, and Ki = 0.8 nM for hBACE2. During Phase I trials, the drug was deemed safe for Alzheimer’s patients with mild cognitive impairment. This drug strongly decreased the amyloid-beta levels and did not result in adverse effects [57]. During Phases II and III, the main goal was to test the efficacy and safety of the drug while ensuring that amyloid-levels were decreasing while cognitive function was simultaneously improving. In 2018, Phase III was terminated due to the decision that the trial was unlikely to meet the primary endpoint [58]. 

Furthermore, Atabecestat was developed by Shionogi and later entered a collaboration with Janssen. Administering Atabecestat, at doses ranging from 5 to 150 mg daily for a duration of up to 14 days in both elderly and young healthy individuals, led to a substantial and consistent decrease of Aβ levels. The reduction was up to 90% in the group receiving 90 mg, observed in both blood plasma and cerebrospinal fluid (CSF) [59]. Phase I studies demonstrated a significant decrease in amyloid-beta deposits in patients in the early stages of Alzheimer’s disease. Phase II trials proved that Atabecestat was able to reduce amyloid-beta levels in both plasma and the cerebrospinal fluid. However, the trial was discontinued in 2018 because of the unfavorable benefit-risk ratio, specifically regarding the increase in liver enzymes and abnormal liver function found in certain patients [57]. 

Elenbecestat is another small-molecule BACE1 inhibitor that was found to decrease amyloid-beta levels in plasma and CSF. A Phase I clinical study (NCT01294540) has revealed that a one-time dose of 50 mg, administered to 73 healthy individuals (both genders, aged 30 to 85 years, across six distinct groups), was both safe and well-received. In an investigation using a single oral dose increasing from 5 to 800 mg, as well as a 14-day multiple ascending-dose study in the range of 25–400 mg, it was established that elenbecestat could significantly lower the Aβ levels in either the blood plasma or cerebrospinal fluid (CSF) by a substantial margin of up to 92%. Specifically, the plasma Aβ(1-X) levels demonstrated a 52% reduction at a 5 mg dose and an impressive 92% decrease at the 800 mg dosage, when compared to baseline levels [60]. A few adverse effects were reported throughout Phases I and II. Elenbecestat was found to delay clinical symptoms of mild dementia in Alzheimer’s patients during a Phase II trial. In June 2018, Biogen declared that the phase II study, which lasted for 18 months, demonstrated not only a substantial decrease in Aβ levels, as verified by amyloid PET imaging, but also less deterioration in cognitive function in patients with a mild to moderate form of Alzheimer’s Disease. Nonetheless, they also reported certain side effects, which included upper respiratory tract infections, unusual dreams and nightmares, contact dermatitis, headaches, diarrhea, and incidents of falling [61]. In 2019, a Phase III trial named MISSION was ended due to the unfavorable benefit-risk ratio [62]. Another drug, CNP520, was terminated in 2020 during Phase III due to safety issues. 

Some therapeutic strategies to increase the efficacy of BACE1 inhibitors are being researched, one of which includes binding BACE1 to a transferrin receptor antibody, which would allow the complex to travel through the BBB better and reduce amyloid-beta levels in the brain [63]. This approach is still being developed and is currently in preclinical phases. The full list of BACE1 inhibitors that are in preclinical/clinical trials is provided below, in Table 3.

### 3.3. The Role of SREBP2 Gene in Cholesterol Regulation Involved in Alzheimer’s Disease

SREBP2 is a transcription factor that regulates the synthesis and uptake of both cholesterol and fatty acids and is the main regulator of cholesterol metabolism [18]. Specifically, SREBP2 activates HMGCR, which is the rate limiting enzyme in cholesterol synthesis [64]. There are about thirty consecutive reactions that contribute to the metabolization of cholesterol, and SREBP2 plays an important role specifically in the mevalonate pathway, which directly links the regulation of cholesterol to amyloid-beta accumulation. During de novo synthesis of cholesterol, SREBP2 binds to SCAP and transfers the complex from the endoplasmic reticulum to the Golgi apparatus, where it is then processed to enter the nucleus and induce transcription for many genes such as HMGCR, mevalonate kinase, and squalene monooxygenase [6]. SREBP2 also binds to the promoter region of ABCA1 and inhibits its transcription, therefore releasing more cholesterol [65]. Multiple signaling pathways can regulate SREBP2 activation during cholesterol synthesis, such as p53, androgen, and Akt, which was a major pathway for the GSK3B target [66]. The p53 pathway can inhibit SREPB2 and reduce the transcription of mevalonate pathway genes (Figure 5A). On the other hand, androgen is a hormone that can enhance SREBP2 activation. Also, the Akt pathway activates SREBP2, which induces the genes involved in cholesterol synthesis (Figure 5B). The Akt pathway is required for the transfer of SREBP2 to the Golgi apparatus. It was found that Akt inhibition leads to the inactivation of SREBP2 and, as a result, cholesterol synthesis is inhibited [18]. Not only is SREBP2 related to the progression of Alzheimer’s disease through the production of cholesterol, it is also directly involved in the regulation of certain genes that contribute to amyloid-beta and tau protein generation. For example, SREBP2 was found to be involved in the regulation of BACE1. SREBP2 activation due to high cholesterol results in the hyperexpression of BACE1. It was determined that blocking SREBP2 expression completely blocked up-regulation of BACE1, which, as previously mentioned, contributes to the production of amyloid-beta [64].

The overproduction of cholesterol has been linked to increase the amyloid-beta production, which offers the potential of SREBP2 to be targeted for Alzheimer’s treatment. One development in SREBP2 inhibition for Alzheimer’s treatment is osmotin. Osmotin is a homolog of adiponectin whose treatment significantly induces AMPK/SIRT1 activation and reduces SREBP2 expression, therefore diminishing amyloidogenic amyloid-beta production [67]. Osmotin treatment has been found to protect against memory impairment, synaptic dysfunction, and neurodegeneration. This treatment also significantly decreased the LDL levels and increased the HDL levels, and in turn also reduced the levels of soluble amyloid-beta in the brain. The use of osmotin also reduces BACE1 expression and alleviates hyperphosphorylation by inhibiting GSK3B through the PI3K/Akt pathway [68]. Although treatment of Alzheimer’s disease using osmotin is still being researched, some small-molecule inhibitors of SREBP2 have been developed and demonstrate promising results in decreasing cholesterol levels. 

Betulin is a drug that inhibits the SREBP2 pathway and decreases the biosynthesis of cholesterol [69]. This drug inhibits SREBP2 by binding to SCAP and promoting the interaction between SCAP and Insig. While statins are typically used to treat hypercholesterolemia, they often activate SREBP2 in the process. Betulin manages to decrease cholesterol levels while also directly inhibiting SREBP2, which makes this drug a promising treatment for Alzheimer’s disease. Although Betulin is often used to treat diabetes and atherosclerosis, the ability of this drug to decrease cholesterol levels and the fact that betulinic acid has been found to prevent Alzheimer’s-induced neurodegenerative issues offer the potential of this SREBP2 inhibitor to benefit Alzheimer’s patients [70]. Other SREBP2 inhibitors, such as Curcumin and Fatostatin, both inhibit SREBP2 activation by binding to SCAP and decreasing cholesterol levels, which demonstrate potential for future Alzheimer’s treatment [70,71]. The full list of SREBP2 inhibitors that are in preclinical/clinical trials is provided below, in Table 4.

### 3.4. The Role of CYP46A1 Gene in Cholesterol Regulation Involved in Alzheimer’s Disease

CYP46A1 is a protein coding gene that is also the rate-limiting enzyme for cholesterol degradation in the brain [72]. Modification of CYP46A1 has been found to improve memory either through isoprenoid synthesis or the production of 24-OH through the modulation of NMDAR [73]. During cholesterol biosynthesis, as illustrated in Figure 1, CYP46A1 leads to the conversion of cholesterol in 24S-OHC, which can cross the blood brain barrier and eliminate cerebral cholesterol. Neurons carry CYP46A1 in order to convert excess cholesterol in the endoplasmic reticulum into 24S-OHC, which forms a major export pathway for excess cholesterol from the brain and metabolizes the excess cholesterol in the liver [6]. Throughout this pathway, the activation of CYP46A1 has been found to increase the expression of SREBP2, which can lead to an increase in cholesterol synthesis [72]. Decreased expression of CYP46A1 in mice was found to increase the amount of cholesterol in neurons, which led to apoptotic death and cognitive deficits [74]. When CYP46A1 is inhibited or mutated, the levels of 24S-OHC significantly decrease, therefore contributing to an accumulation of cholesterol in the brain since 24S-OHC is not available to extrude excess cholesterol from the brain. Studies have found that CYP46A1 activation helps reduce cholesterol levels in the brain, not by slowing the rate of synthesis of cholesterol but instead by increasing the rate at which cholesterol is expelled from the brain. CYP46A1 activation also leads to a decreased production of both amyloid-beta and tau proteins [73]. The downregulation of CYP46A1 was found to increase amyloid-beta levels through the recruitment of APP in lipid rafts and eventually resulted in neuronal death [74]. Studies indicate that a few gene variations in CYP46A1, including the CYP46A1 T allele, are possible genetic risk factors of Alzheimer’s disease [75]. Due to its role in amyloid-beta and 24S-OHC production, as well as its possible involvement as a genetic risk factor, the activation of CYP46A1 has the potential to treat Alzheimer’s disease.

The most successful and advanced drug for Alzheimer’s disease that activates CYP46A1 is Efavirenz. This drug is an anti-HIV medication that is used to treat patients with mild cognitive impairment or early dementia due to Alzheimer’s disease [76]. The activation of CYP46A1 due to Efavirenz resulted in the enhancement of brain cholesterol turnover and behavior improvements. Efavirenz interacts with CYP46A1 by binding to the P450 active site and activating it. Studies have found that this drug successfully reduces the amyloid-beta levels, increases cholesterol elimination, and improves cognitive behavior. Small doses of Efavirenz were found to be safe in Alzheimer’s patients and even have some advantages over other potential options for Alzheimer’s treatment. For example, Efavirenz increases the production of 24S-OHC and promotes the elimination of cellular cholesterol [77]. Also, CYP46A1 activation will only affect the central nervous system, whereas synthetic LXR agonists may affect multiple systems of the body (more detail to be given in the next section). 

Some other drugs that target CYP46A1 for Alzheimer’s treatment are huperzine and galantamine. Galantamine found modest but consistent cognitive benefits over a short period of time and has the potential to treat Alzheimer’s throughout long-term treatment [78]. Huperzine A is a Chinese herb extract that has been found to improve cognitive function in participants with Alzheimer’s disease, but some issues were reported throughout trials regarding poor methodological quality [79]. Studies have also found that adeno-associated virus gene therapy can be used to overexpress CYP46A1 in order to increase 24S-OHC in the brain [80]. Using this form of therapy, CYP46A1 overexpression before the onset of amyloid plaques aided in the improvement of cognitive function in mice, as well as the reduction of microgliosis and astrogliosis. However, further studies need to be done to determine the long-term effects of selective CYP46A1 overexpression. The full list of CYP46A1 inducers that are in preclinical/clinical trials is provided below, in Table 5.

### 3.5. The Role of ABAC1 Gene in Cholesterol Regulation Involved in Alzheimer’s Disease

ABCA1 is an ATP binding cassette that mediates cellular cholesterol efflux (Figure 6). ABCA1 is expressed with the highest levels in liver hepatocytes but is also found in the central nervous system through the neurons, astrocytes, and microglia [81]. It is an integral membrane protein that utilizes ATP to transport cholesterol to APP, and its main function is to maintain lipid homeostasis by controlling the transport of cholesterol [82]. ABCA1 also interacts with ApoE1 to develop HDL, which aids in the efflux of cholesterol from the periphery to the liver. ABCA1 may also protect cells from the cytotoxic effects of excess cholesterol. ABCA1 plays a vital role in central nervous system cholesterol transport through the reverse cholesterol transport pathway. ApoE is mainly synthesized in astrocytes, where ApoE then receives cholesterol and phospholipids from ABCA1. The cholesterol and phospholipids are then secreted into the brain as lipoproteins, and excess cholesterol is internalized by LRP1. It is believed that CYP46A1 utilizes ABCA1 to convert cholesterol into 24-OHC and consequently remove cholesterol from the brain [83]. ABCA1 is involved in many cell-signaling pathways, but two pathways in particular are cholesterol-dependent. Through de novo synthesis, cholesterol in cells can be converted into oxysterols. Oxysterols increase ABCA1 expression through the activation of the liver X receptor pathway, which then forms a complex with retinoid X receptor (RXR) and together form a transcription factor that binds to ABCA1 to increase its expression [82]. The LXR/RXR pathway can be activated to increase ABCA1 expression, which would increase the efflux of cholesterol out of the brain by inversely decreasing BACE1 expression. By activating this pathway, ABCA1 can be activated and BACE1 can be inhibited, which decreases the cholesterol and amyloid-beta levels in the brain [84]. The other cholesterol-dependent pathway that ABCA1 is involved in is the SREBP2 pathway. Unlike the LXR/RXR pathway explained above, the activation of the SREBP2 pathway decreases the expression of ABCA1. As mentioned in previous sections, this pathway is activated by the binding of SCAP to SREBP2, which promotes the transcription of many genes including SREBP2 itself. MiR-33a is co-transcribed with SREBP2, which has been found to inhibit ABCA1 expression. Therefore, activation of the LXR/RXR or inactivation of the SREBP2 pathway by inhibiting miR-33a both offer possible therapeutic possibilities for Alzheimer’s disease, by increasing the expression of ABCA1.

The activation of ABCA1 has been the target for multiple studies involved in Alzheimer’s treatment and has been found to offer positive results in decreasing cholesterol levels. Synthetic LXR agonists such as T0901317 have been proven to increase ABCA1 expression while simultaneously reducing the soluble amyloid-beta levels in the brain [85]. One study with mice even found that T0901317 reversed memory deficits. Synthetic LXR agonists have also been found to stimulate the enzymatic degradation of amyloid-beta by microglia. Synthetic RXR agonists such as bexarotene have more conflicting results, but studies have found this type of treatment to enhance the clearance of amyloid-beta plaques and improve cognitive performance. However, both LXR and RXR agonists led to unfavorable effects in the liver and therefore few developments have reached clinical trials. Small molecule inducers have been developed to enhance ABCA1 expression while avoiding the hepatoxic side effects of LXR agonists. Antagonists of P2X7, such as AZ-1 and AZ-2, resulted in the enhancement of ABAC1 enhancement and activity, which significantly increased cholesterol efflux [86]. These compounds were also found to activate the LXR pathway indirectly. Also, AZ-1 and AZ-2 significantly increased ABAC1 protein levels in CNS cells such as astrocytes, microglia, and vascular pericytes. Other nonlipogenic ABCA1 inducing compounds have been researched to selectively activate ABAC1 while keeping the liver unaffected. A new small molecule ABCA1 inducer called E17241 dose-dependently upregulates ABCA1 expression and increases cholesterol efflux [87]. Administered at two different doses, this drug enhances ABAC1 expression without provoking any adverse effects on the liver. Compared to the other therapeutic developments mentioned above, E17241 offers antiatherogenic effects upon oral administration without the need for nanoparticles. The full list of ABCA1 inducers that are in preclinical/clinical trials is provided below, in Table 6.

### 3.6. The Role of TREM2 Gene in Cholesterol Regulation Involved in Alzheimer’s Disease

TREM2 is a transmembrane immune receptor that is expressed by microglia in the brain and enhances the rate of phagocytosis. This gene also has anti-inflammatory properties and modulates inflammatory signaling [88]. In the microglia, signaling of the TREM2/DAP12 pathway was found to promote cholesterol efflux and therefore reduce how much intracellular cholesterol is stored as cholesterol esters (Figure 7). Extracellular ligand binding activates TREM2, which then activates DAP12 [89]. ITAM then becomes phosphorylated, which causes the Syk kinase to activate downstream signaling molecules including PI3K, ERK, PLCy, and Vav [90]. The cholesterol is then transported out of the microglia via ApoE-containing lipoproteins. Phosphorylation of DAP12 often affects other pathways and has the ability to inhibit the RAS/MEK/ERK pathway. This inhibition may decrease the secretion of pro-inflammatory cytokines. The phosphorylation of DAP12 also has a direct effect on the uptake of amyloid-beta through its involvement with SHIP1 and the SHIP1/CD2A/RIN3/BIN1 complex. TREM2 also mediates proliferation and survival of myeloid cells by the alteration of the Wnt/B-catenin pathway. Studies found that TREM2 stabilized B-catenin by inhibiting its degradation through the Akt/GSK3B pathway [91]. TREM2 is also involved in mTOR signaling through the same pathway, which has the ability to alter microglial survival. In relation to Alzheimer’s disease, studies have found TREM2 expression leads to an increase in tau phosphorylation, but this relation has not been studied as much as TREM’s connection to amyloid-beta. TREM2 is thought to have a time-dependent effect on amyloid-beta accumulation. The signaling of TREM2 was found to increase amyloid-beta accumulation in early stages of Alzheimer’s disease and decrease amyloid-beta accumulation in later stages [92]. Other studies have found TREM2 overexpression to reduce amyloid-beta accumulation [93]. Through the activation of the NFκB pathway, BACE1 expression increases and TREM2 expression decreases due to regulation of miRNA-34a, which causes amyloid-beta accumulation [94]. Mutations of TREM2 contribute to late-onset Alzheimer’s disease because these variants disturb TREM2 signaling and its functions [88].

The increased expression of TREM2 in Alzheimer’s patients has been researched as a possible therapeutic strategy. Most strategies to treat Alzheimer’s disease by targeting TREM2 found it most effective to stimulate TREM2 in the early stages of the disease. The soluble form of TREM2 (sTREM2) has reached preclinical trials due to its involvement in microglial regulation and tau phosphorylation levels [95]. Soluble TREM2 is released into the extracellular space when TREM2 is cleaved and has been found to promote cell survival. Increased levels of sTREM2 are found in the cerebrospinal fluid of those with Alzheimer’s disease. Studies found that an increase in sTREM2 leads to a decrease in amyloid-beta levels as well as diminished cognitive decline [96]. Although sTREM2 drugs remain in preclinical trials, other therapeutic strategies such as anti-TREM2 antibodies have reached clinical trials. AL002 is an anti-human TREM2 antibody that binds to TREM2 and activates signaling. It was found to cause microglia to express more pro-inflammatory and repair genes, while significantly decreasing amyloid-beta accumulation [97]. Phase 1 clinical trials proved that AL002 was able to reduce sTREM2 levels in the brain without causing any adverse effects. A Phase 2 study is currently recruiting participants and is expected to be completed in January 2024 [98]. 

On the other hand, AL003 is a monoclonal antibody that targets the receptor CD33 and as a result inhibits TREM2. Phase 1 clinical trials demonstrated conflicting results from this drug. Participants from the multiple-ascending-dose phase demonstrated an increase in soluble CD33 levels in Cerebrospinal fluid (CSF) with no adverse effects. During the phase of single ascending doses, 38 healthy individuals were given infusions with increasing dosages from one of eight levels. The evaluation parameters included safety, tolerability, pharmacokinetics, pharmacodynamics, and immunogenicity. It was observed that in healthy volunteers, the drug was well received up to a dosage of 15 mg/kg. However, one participant developed hip inflammation while another experienced severe hypersensitivity, and both were hospitalized [99]. In June 2022, AL003 was terminated without any justified reason. Another agnostic antibody drug that has been developed is TREM2 TVD-Ig, which is an agnostic antibody that activates TREM2. This drug aims at enhancing microglia functions and reducing amyloid pathology [100]. During preclinical development, this drug was found to enhance microglial phagocytosis of amyloid plaques. Future experiments on this drug aim at focusing on the microglia activation phenotype, which is necessary to understand the mechanism of this drug [101]. An antibody transport vehicle named DNL919 was developed to activate TREM2 and analyze the effects this drug has on CSF1R [102]. This drug was put on hold by the FDA in January 2022 but is currently recruiting participants and is set to finish Phase 1 clinical trials in July 2023 [103]. The full list of TREM2 inducers that are in preclinical/clinical trials is provided below, in Table 7.

## 4. Discussion

### 4.1. Protein Targets for Further Investigation

Each protein that was discussed in Section 3 offers a unique advantage to treat Alzheimer’s disease by regulating the metabolism of cholesterol. GSK3B’s involvement in the Wnt/B-catenin and Akt pathways, as well as its role in activating SREBP2, makes it an attractive target since it affects amyloid-beta production, tau phosphorylation, and cholesterol synthesis. GSK3B inhibitors also have the most progress with preclinical and clinical trials, but many of these drugs are used to treat cancer rather than Alzheimer’s disease. BACE1 is involved in two different pathways that produce amyloid-beta through reverse cholesterol transport, and its role in APP cleavage offers potential to treat Alzheimer’s disease early on rather than treating the disease later in its progression. BACE1 inhibitors usually consist of small-molecule inhibitors that aim to reduce amyloid-beta levels in the cerebrospinal fluid. Although these drugs have made substantial progress by reaching Phase 3 of clinical trials, they have all been terminated due to safety and effectiveness issues. SREBP2 plays an important role in both cholesterol synthesis and amyloid-beta production during de novo synthesis. SREBP2 has a direct relation to other proteins involved in cholesterol synthesis, and its involvement in multiple different pathways offers alternative routes to inhibit the protein. Different SREBP2 inhibitor drugs have been developed, but many of them still remain in the preclinical phase. CYP46A1 is a key gene in the removal of cholesterol from the brain. Its activation has been found to reduce cholesterol, amyloid-beta, and tau protein levels in the brain. Not many CYP46A1 inducers have been developed, but those that are developed have been found to be effective in removing cholesterol from the brain. ABCA1 is critical in the transport of cholesterol out of the brain, and its involvement in both the LXR/RXR and SREBP2 pathways offer different ways to activate the gene and remove cholesterol. While quite a few ABAC1 inducers have been developed, most of them remain in the preclinical phase. Only one drug in particular is currently in Phase 3 of clinical trials; TREM2’s involvement in the TREM2/DAP12 pathway promotes cholesterol efflux, while its involvement in the Wnt/B-catenin and Akt pathways promote cell survival. TREM2 target drugs are still early in their development, but their ability to alter TREM2 in early stages of Alzheimer’s disease has been found to promote cell survival and decrease amyloid levels in the brain.

For the perspective of drug discovery, especially small molecule inhibitor discovery, the protein targets for inhibition intervention are preferred. With known crystal structures, chemical compounds can be evaluated for their binding affinities to the active sites of the protein targets. Various computational platforms have been developed for this purpose [104,105]. Identifying small molecule inhibitors is more straightforward than designing activators for target proteins. Therefore, ABCA1, CYP46A1, and TREM2 were ranked the lowest because they must be activated rather than inhibited in order to prevent the accumulation of amyloid-beta plaques, tau phosphorylation, and increasing cholesterol levels. In addition, while ABCA1 has many clinical trials done, most were terminated. With the consideration of available crystal structures and the trial and experiments done on each of the other three targets, the top protein target is SREBP2. It has a limited number of clinical trials and a well-established crystal structure. It is involved in two different pathways and interacts with two other target genes. The next ranked target is BACE1. This protein has a well-established crystal structure and has a limited number of clinical trials done. It is also involved in three different pathways and has interactions with three other target genes involved in Alzheimer’s disease and cholesterol, which allows for multiple ways to inhibit not only BACE1 but other target genes associated with cholesterol production and Alzheimer’s disease. The next ranked gene is GSK3B. GSK3B has a well-established crystal structure and is involved in two different pathways, one being the Wnt/B-Catenin pathway. This gene is not ranked higher because it has the highest number of clinical trials and only interacts with one other target gene. Compared to SREBP2, more research has been conducted for BACE1 and GSK3B on inhibitor identification. This justifies that more space can be explored for inhibitors for SREBP2.

### 4.2. Potential Approaches for Identifying Inhibitors for Selected Protein Targets

As for each of the protein targets for inhibition, (i.e., SREBP2, BACE1 and GSK3B), the interaction between potential ligands and the target protein can be first studied via in silico approaches, to accelerate the pace for compound discovery and reduce the experimental cost. While there are various programs available for identifying small molecule inhibitors for a selected protein target, Molsoft ICM Pro outperforms other programs, such as Autodock, DOCK, FlexX, Gold, FITTED, and MOE, in its flexible docking and covalent docking [105]. In addition, Molsoft ICM pro won the docking pose and energy prediction in drug design data resource (D3R) challenge for both 2017 and 2018 [104]. Specifically, Molsoft ICM-Pro is a ligand-protein docking program that can analyze and determine the specific position and orientation of a ligand bound to the protein and, overall, will identify the binding posture between the ligand and the target protein with the lowest binding energy (i.e., the best binding affinity). On the basis of the crystal structure of the selected protein, which is typically downloaded from the PDB database, the program identifies the ligand binding sites while also screening virtual libraries of compounds, and lastly evaluates and ranks the drug targets based on their interactions. The binding affinity data can be further used to build models to predict the quantitative structure-activity relationship (QSAR) [106,107,108] for compound screening. ICM-based pipelines have been developed to identify inhibitors to inhibit growth of foodborne pathogens [109,110], inhibit a protease essential for SARS-CoV-2 replication [111], and inhibit a protein involved in the progression of Alzheimer’s disease [112]. Once the relationship between the structures have been determined, the effectiveness and safety of the inhibitors with good binding affinities can be analyzed in preclinical studies. Preclinical studies will be used to evaluate which candidate has the best pharmacokinetic parameters in humans based on safety, toxicity, and efficacy determination [113]. After the preclinical studies have deemed the target drugs as safe and efficient, they will move on to the clinical trials. Phase I will determine whether the drug is safe and tolerable for humans by using a small group of healthy participants. Phase II will be performed on a larger group of participants and helps establish the necessary doses for the drug as well as the drug’s efficacy. Phase III will evaluate the long-term safety of the drug by performing randomized controlled trials within a large group of participants over a 6–12-month period. Phase IV will most likely be post-market surveillance trials to observe the safety, effectiveness, and economic outcomes [114,115].

### 4.3. Limitation of This Work

While extensive research has been conducted in the last few decades on Alzheimer’s disease, only seven drugs have been approved by FDA to treat Alzheimer’s disease. One reason for this is that the progression of Alzheimer’s disease is regulated by a large number of genes, reactions, and pathways which are still under investigation. This work does not aim to illustrate the detail of all genes/pathways known for their involvement in Alzheimer’s disease. Interested readers can refer to review papers (e.g., [116,117,118,119,120]) for the intricacies of the mechanism. In addition, interested readers can find the review of existing treatments/drugs for Alzheimer’s disease in the literature (e.g., [121,122,123,124,125,126,127,128]). Different from these existing reviews, this work conducted a protein-protein interaction analysis to identify top protein targets from existing clinical data (encoded in the DisGeNET knowledge platform for disease genomics). While there are other genes playing an important role in regulating the progression of Alzheimer’s disease, only six genes are focused on in this work, due to space constraints. The key reactions/pathways for those genes were then given to illustrate the involvement of these genes in Alzheimer’s disease regulation. The existing preclinical or clinical trials were provided for each of these genes. The recommendation of top drug targets in this work may have limitations due to the small number of genes that were investigated in this work. A similar investigation could be conducted for more genes in the future.

As for the intervention to the protein targets, this work mainly focuses on recommendations for the targets for inhibition, instead of those that needed activation. One reason is that computational tools are available to screen millions of compounds for potential inhibitors in computers. This is also because the team has extensive experience in systems biology and drug screening but limited knowledge on protein activation approaches. A more thorough investigation on protein targets needed for activation for Alzheimer’s disease intervention is strongly recommended in future research.

## 5. Conclusions

High levels of cholesterol in the brain have been attributed to the progression of Alzheimer’s disease due to the connection between cholesterol synthesis and amyloid-beta plaques and tau-phosphorylation. Recently, studies have focused on targeting specific proteins involved in cholesterol synthesis/transport in order to treat Alzheimer’s disease. By utilizing existing clinical date from DisGeNET and undergoing protein-protein network analysis on cholesterol metabolism and Alzheimer’s disease, the top target genes selected were GSK3B, BACE1, SREBP2, CYP46A1, ABCA1, and TREM2. Each of these proteins were involved with cholesterol synthesis/transport, as well as the production of amyloid-beta and/or tau proteins. SREBP2, BACE1, and GSK3B were selected as the preferred target proteins for inhibition, while ABCA1, CYP46A1, and TREM2 are the protein targets for activation. The reactions and pathways for each of these targets in regulating brain cholesterol for Alzheimer’s disease progression were thoroughly investigated in this work. SREBP2 was recommended as a good target for future small molecule inhibitor screening due to its well-established crystal structure and its limited number of clinical trials. Molsoft ICM pro was recommended for virtual screening of chemical compounds with good binding affinities to the selected targets. The results from this work provide the Alzheimer’s research community with an overview of protein targets along with potential targets and approaches for further investigation.

## Figures and Tables

**Figure 1 ijerph-20-06217-f001:**
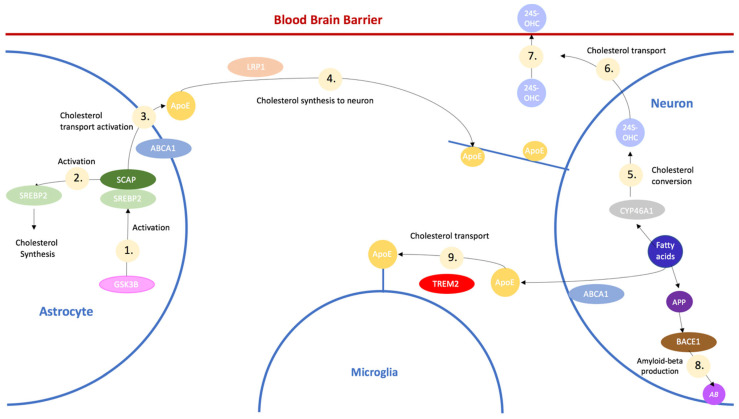
Overview of the main pathways involved in cholesterol transport involved in Alzheimer’s Disease. Although not directly illustrated, SREBP2 is activated by GSK3B through the Akt pathway. De novo synthesis is indicated by Pathway 1, where SREBP2 and SCAP bind to each other and transport cholesterol from the ER to the Golgi apparatus. The SREBP2/SCAP complex activates SREBP2, which is involved in cholesterol synthesis in Pathway 2. Through Pathway 3, ABCA1 transports cholesterol to ApoE, which then transports the cholesterol to LRP1 to become synthesized in the neuron by Pathway 4. Pathways 5 and 6 demonstrate CYP46A1 converting excess cholesterol into 24S-OHC and then being exported from the neuron, which then crosses the Blood Brain Barrier through Pathway 7. Through Pathway 8, BACE1 cleaves APP to produce amyloid-beta plaques. When cholesterol is transported to the microglia via Pathway 9, TREM2 aids in cholesterol efflux out of the brain.

**Figure 2 ijerph-20-06217-f002:**
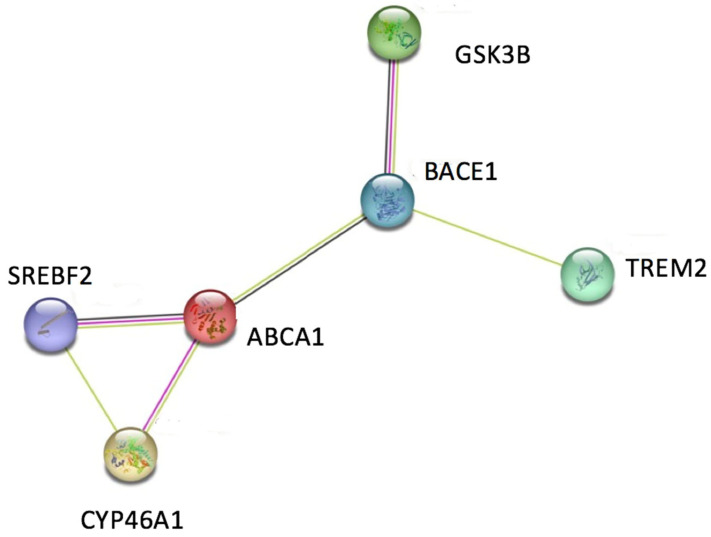
Overview of network between top genes associated with brain cholesterol metabolism and Alzheimer’s Disease (produced by the STRING program). Each colored line represents an individual known or predicted interaction between the gene pair. More lines indicate more interactions between the two genes they are connected to. Filled nodes indicate some known or predicted crystal structure that exists for the gene. Genes with a greater filled node indicate that more crystal structures exist for that specific gene.

**Figure 3 ijerph-20-06217-f003:**
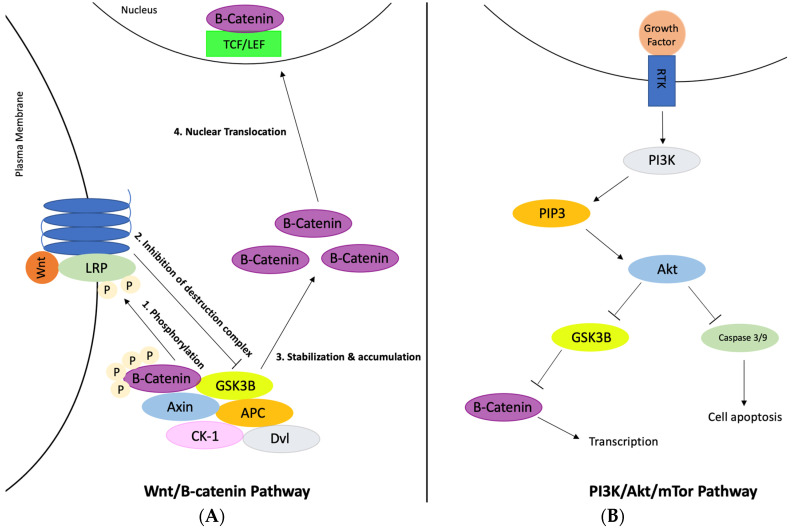
The major pathways and interactions regulated by GSK3B: (**A**) Wnt/B-catenin pathway and (**B**)PI3K/Akt/mTor pathway. In the Wnt/B-catenin pathway, the binding of Wnt to LRP causes the GSK3B complex with other proteins to be inhibited. This same complex phosphorylates the LRP/Wnt/ligand complex. The inhibited GSK3B complex stabilizes this pathway and causes the accumulation of B-catenin outside of the plasma membrane. B-catenin then translocates into the nucleus and forms a complex with the TCF/LEF transcription factors, which reduces transcription and DNA binding. In the PI3K/Akt/mTor pathway, the binding of the growth factor to the receptor tyrosine kinase stimulates PI3K and PIP3 activity, which then stimulates Akt activity. This activation leads to the phosphorylation of GSK3B, which inactivates the protein. Once GSK3B is inhibited, the assembly of B-catenin is regulated for the Wnt signaling that is important to maintain synaptic signaling.

**Figure 4 ijerph-20-06217-f004:**
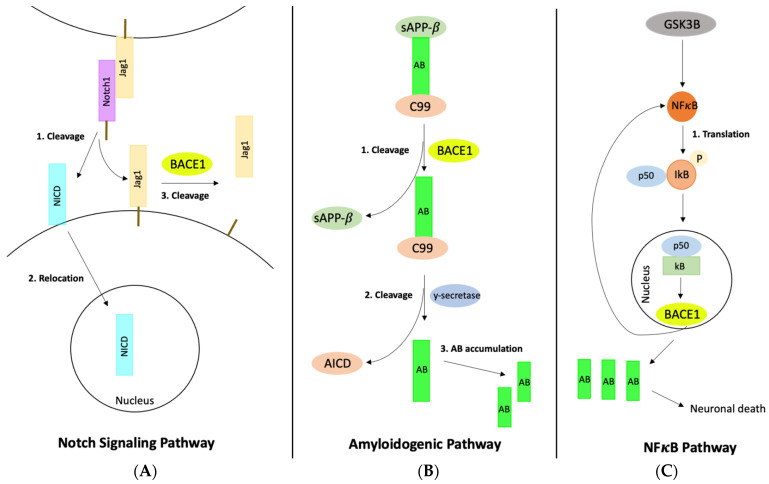
The major pathways and interactions regulated by BACE1: (**A**) Notch signaling pathway, (**B**) Amyloidogenic pathway, and (**C**) NFκB pathway. In the Notch signaling pathway, the binding of Notch1 to the Jag1 ligand causes a conformational change in Notch1 and results in cleavage. This cleavage releases NCID, which then relocates into the nucleus. NCID’s relocation activates downstream signaling. This signaling can be regulated by binding Jag1 to the surface of the cell. BACE1 then cleaves Jag1, which regulates the activity of Jag1 in the sending cell and stimulates Notch1 signaling in the receiving cell. In the amyloidogenic pathway, APP is cleaved by BACE1 to generate amyloid-beta. The two cleavage fragments, C99 and sAPP-β, are released. The C99 complex with amyloid-beta is cleaved by γ -secretase, which generates AICD and releases amyloid-beta. The BACE1 cleavage is the rate-limiting step in amyloid-beta formation, therefore the liberation of amyloid-beta leads to an accumulation of amyloid-beta. In the NFκB pathway, GSK3B activates NFκB, which causes the phosphorylation of IκB. This phosphorylation causes IκB to relocate to the nucleus and bind to DNA. This activates multiple target genes associated with amyloid-beta production including BACE1, which can reactivate NFκB.

**Figure 5 ijerph-20-06217-f005:**
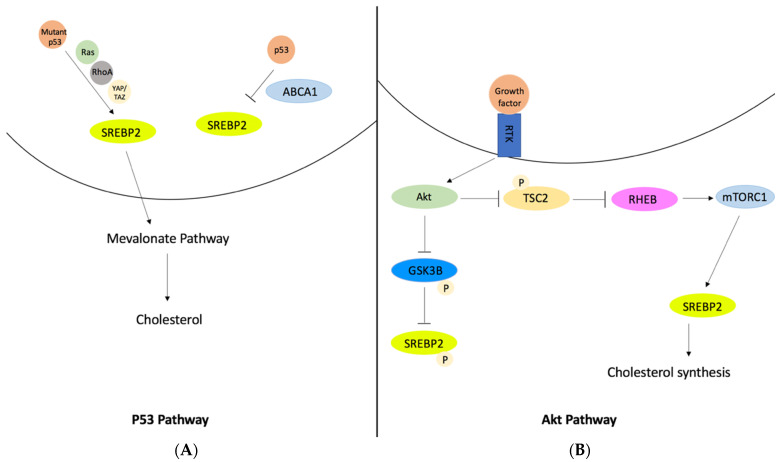
Major signaling pathways that regulate SREBP2 activation during cholesterol synthesis: (**A**) P53 pathway, and (**B**) Akt pathway. In the p53 pathway, mutant p53 binds to SREBP2 and subsequently activates the Ras, RhoA, YAP/TAZ pathways, while p53 blocks the activation of SREBP2 through ABCA1. Once SREBP2 is activated, it signals the mevalonate pathway to synthesize acetyl-CoA into cholesterol. In the Akt pathway, growth factors bind to receptor tyrosine kinases to activate PI3K, which then activates Akt. Akt phosphorylates and inhibits TSC2, which inhibits RHEB and activates mTORC1. MTORC1 works with other targets to stimulate and activate SREBP2. However, the activation of Akt may directly phosphorylate GSK3B, which will mediate phosphorylation and inhibition of transcription factors including SREBP2. This promotes proteasomal degradation.

**Figure 6 ijerph-20-06217-f006:**
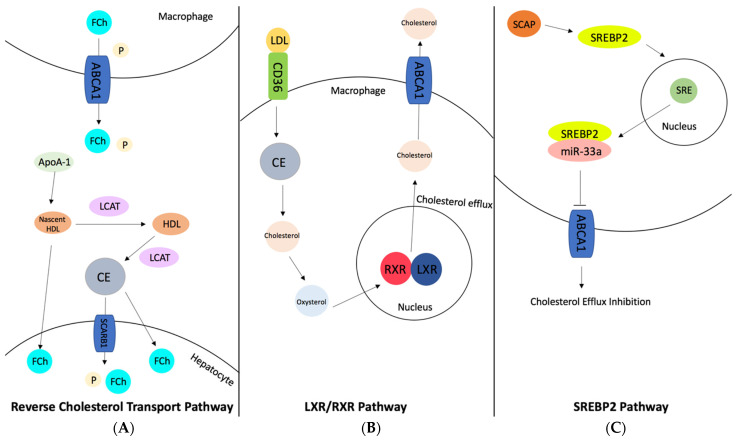
Major pathways in which ABCA1 plays an important role for cholesterol transport in brains: (**A**) Reverse cholesterol transport pathway, (**B**) LXR/RXR pathway, and (**C**) SREBP2 pathway. In the reverse cholesterol transport pathway, free cholesterol is transported out of the macrophage to ApoA-1. ApoA-1 then becomes pre-B-HDL or nascent HDL. Nascent HDL then becomes mature HDL which is mediated by LCAT. LCAT then mediates cholesterol esterification of HDL into cholesterol ester (CE). The cholesterol esters are then taken up by SCARB1 receptors and taken up into hepatocytes. Cholesterol esters are then hydrolyzed to generate free cholesterol (FCh) in hepatocytes. In the LXR/RXR pathway, ligand LDL binds to receptor CD36, which oxidizes LDL into cholesterol esters. The cholesterol esters are converted into cholesterol, which is then converted into oxysterols. The oxysterols activate LXR/RXR, which then promotes cholesterol efflux from the nucleus, which then activates ABCA1 to transfer cholesterol out of the macrophage to Apo acceptors. In the SREBP2 pathway, SCAP transports SREBP2 from the ER to the Golgi apparatus. SREBP2 is then translocated to the sterol regulatory element in the nucleus. Then miR-33a is co-transcribed with SREBP2, which inhibits ABCA1 activity. By inhibiting ABCA1, cholesterol efflux is hindered, therefore cholesterol accumulates. However, inhibiting miR-33a would activate ABCA1 and promote cholesterol efflux.

**Figure 7 ijerph-20-06217-f007:**
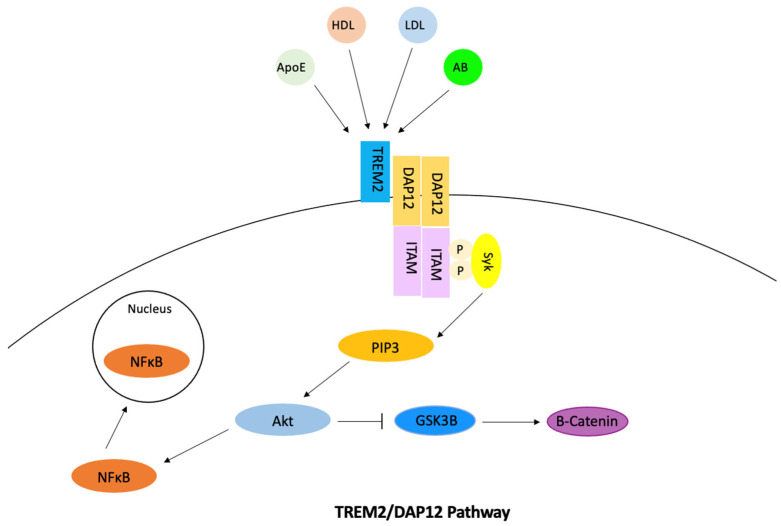
The TREM2/DAP12 pathway. In this pathway, various ligands bind to TREM2 which creates an electrostatic interaction between TREM2 and DAP12. This interaction generates tyrosine phosphorylation of DAP12 with ITAM. This recruits the Syk kinase to activate signaling molecules such as PIP3, which then activates Akt. Through the Akt pathway, GSK3B is inhibited and therefore B-catenin is stabilized. The activation of Akt also activates NFκB, and NFκ B then relocates into the nucleus and binds to DNA. This increases the release of proinflammatory cytokines into the cytoplasm.

**Table 1 ijerph-20-06217-t001:** The target genes and the characteristics of their crystal structure found on the PDB Database. *BACE1 has around 225 crystal structures with unique PDB codes, therefore only the first 10 were included.

Gene	PDB Code	Method	Resolution
ABCA1	6S4N	X-Ray Diffraction	1.9 Å
	6S4K	X-Ray Diffraction	1.6 Å
	6S4T	X-Ray Diffraction	2.0 Å
CYP46A1	3MDM	X-Ray Diffraction	1.6 Å
	7LRL	X-Ray Diffraction	1.995 Å
	7N3M	X-Ray Diffraction	1.698 Å
	2Q9F	X-Ray Diffraction	1.9 Å
	7N3L	X-Ray Diffraction	1.631 Å
	7N6F	X-Ray Diffraction	1.4 Å
*BACE1	7N4N	X-Ray Diffraction	1.41 Å
	6EJ2	X-Ray Diffraction	1.46 Å
	7D5B	X-Ray Diffraction	1.31 Å
	3VF3	X-Ray Diffraction	1.48 Å
	5HDZ	X-Ray Diffraction	1.49 Å
	6UWP	X-Ray Diffraction	1.29 Å
	6UWV	X-Ray Diffraction	1.47 Å
	4L7G	X-Ray Diffraction	1.38 Å
	6EQM	X-Ray Diffraction	1.35 Å
	7MYI	X-Ray Diffraction	1.25 Å
TREM2	5UD8	X-Ray Diffraction	1.8 Å
	6XDS	X-Ray Diffraction	1.466 Å
GSK3B	1J1B	X-Ray Diffraction	1.8 Å
	1Q5K	X-Ray Diffraction	1.94 Å
	4AFJ	X-Ray Diffraction	1.98 Å
	2JDO	X-Ray Diffraction	1.8 Å
	2 × 39	X-Ray Diffraction	1.93 Å
	1O6K	X-Ray Diffraction	1.7 Å
	1O6L	X-Ray Diffraction	1.6 Å
SREBP2	1UKL	X-Ray Diffraction	3.0 Å

**Table 2 ijerph-20-06217-t002:** GSK3B inhibitor drugs in preclinical or clinical trials along with their progress [49].

Drug Name	Type	Clinical Trials	Status
Tideglusib	Thiadiazolidinone	Phase 2b	Terminated, missed primary endpoint
AL001	Monoclonal antibody	Phase 3	Active
SAR502250	Pyridine/pyrimidine	Preclinical	N/A
Laduviglusib HCl	Hydrochloride of CHIR-99021	Preclinical	N/A
SB-216763	Maleimide	Preclinical	Terminated, not selective
AT7519	Multi-CDK inhibitor	Phase 1	Active
TWS119	Pyridine/pyrimidine	Phase 1	Active
SB415286	Maleimide	Preclinical	N/A
AZD1080	Pyridine/pyrimidine	Phase 2	Terminated, induced severe histopathological changes in gall bladder
AR-A014418	Pyridine/pyrimidine	Preclinical	N/A
TDZD-8	Thiadiazolidinone	Preclinical	N/A
LY2090314	ATP-competitive inhibitor	Phase 2	Terminated, slow enrollment and safety issues
BRD0705	Pyrazolodihydropyridine	Preclinical	N/A
Alsterpaullone	Paullone	N/A	N/A
IM-12	Indolylmaleimide	Phase 2	Completed for chronic disease of the graft against the host
1-Azakenpaullone	Paullone		
Indirubin-3′-oxime	Indole	Phase 4	Completed for childhood acute promyelocytic leukemia
Resibufogenin	Bufadienolide toxin	Phase 2	Completed for pancreatic cancer
CP21R7	Selective GSK3B inhibitor	N/A	N/A
PF-04802367	Oxazole-carboxamide	Preclinical	N/A

**Table 3 ijerph-20-06217-t003:** BACE1 inhibitor drugs in preclinical or clinical trials along with their progress [57].

Drug Name	Type	Clinical Trials	Status
Verubecestat	Small-molecule inhibitor	Phase 3	Terminated
Lanabecestat	Small-molecule inhibitor	Phase 3	Terminated, not likely to meet primary endpoint
Atabecestat	Brain-penetrable small-molecule inhibitor	Phase 3	Terminated, benefit-risk ratio not ideal due to elevations in liver enzymes
Elenbecestat	Small-molecule inhibitor	Phase 3	Terminated, unfavorable benefit-risk ratio
CNP520	Small-molecule inhibitor	Phase 3	Terminated, safety issues

**Table 4 ijerph-20-06217-t004:** SREBP2 inhibitor drugs in preclinical or clinical trials along with their progress [66].

Drug	Type	Clinical Trials	Status
Fatostatin	Non-sterol diarylthazole derivative	Preclinical	N/A
Alpha-Tocotrienol	Vitamin E component	Phase 3	Completed
Artesunate	Antimalarial injection	FDA approved for malaria, Preclinical for Alzheimer’s	N/A
Emodin	Naturally occurring herbal ingredient	Preclinical	Terminated trial for kidney disease due to high rate of drop-out
miRNA-185/342, miRNA-33a	Regulatory, silencing RNAs	Preclinical	N/A
Osmotin	Antifungal protein	Preclinical	N/A
Betulin	Small-molecule inhibitor	Preclinical	N/A
Curcumin	Phytopolyphenol pigment	Phase 2	Completed

**Table 5 ijerph-20-06217-t005:** CYP46A1 inducer drugs in preclinical or clinical trials along with their progress.

Drug	Type	Clinical Trials	Status
Efavirenz	Non-nucleoside reverse transcriptase inhibitor	Phase 1 for Alzheimer’s	Completed, FDA approved for HIV/AIDS
Huperzine A	Chinese herb extract	Phase 2	Completed
Galantamine Hydrobromide	Cholinesterase inhibitor	FDA approved	Sold under name Razadyne for mild-to-moderate stages

**Table 6 ijerph-20-06217-t006:** ABCA1 inducer drugs in preclinical or clinical trials along with their progress.

Drug	Type	Clinical Trials	Status
T0901317	LXR agonist	Preclinical	N/A
GW3965	LXR agonist	Preclinical	N/A
LXR-623	LXR agonist	Phase 1	Terminated, associated with neurological and psychiatric side effects
AZ-10606120	P2X7 blocking molecules	Preclinical	N/A
AZD9056	P2X7 inhibitor	Phase 2b	Terminated, failed to demonstrate efficacy
CL2-57	LXR agonist	Preclinical	N/A
E17241	Small-molecule ABCA1 upregulator	Preclinical	N/A
BMS-852927	LXR agonist	Phase 1	Terminated, increased peripheral side effects such as neutropenia
CS-6523	Small-molecule ApoE mimetic, ABCA1 agonist	Preclinical	N/A
ETC-216	Reconstituted HDL nanoparticles	Phase 2	Terminated, adverse effects
CSL-111	Reconstituted HDL nanoparticles	Phase 2	Terminated, adverse effects
CSL-112	Reconstituted HDL nanoparticles	Phase 3	Active

**Table 7 ijerph-20-06217-t007:** TREM2 inducer and inhibitor drugs in preclinical or clinical trials along with their progress.

Drug	Type	Clinical Trials	Status
AL002	Monoclonal antibody	Phase 2	Recruiting
AL003	Monoclonal antibody	Phase 1	Completed but later terminated
DNL919	Antibody transport vehicle	Phase 1	Recruiting
sTREM2	Soluble form of TREM2	Preclinical	N/A
TREM2 TVD-Ig	Tetra-variable domain antibody	Preclinical	N/A

## Data Availability

Data can be provided upon request.

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
