# Peer review of "Investigation of Potential Drug Targets for Cholesterol Regulation to Treat Alzheimer’s Disease"

_ijerph, 2023, doi:10.3390/ijerph20136217_

Round 1

Reviewer 1 Report

The manuscript, in general, is fine however, the Introduction needs more references for statements that are made. Figures, especially Figure 2 need to be improved. In the text it needs to made more clear if data refer to humans or animal experiments.

Author Response

The manuscript, in general, is fine however, the Introduction needs more references for statements that are made.

Response: Thanks for the comments. We have added more references in the Introduction section. Please refer to that section for the changes we have made.

Figures, especially Figure 2, need to be improved. In the text it needs to made more clear if data refer to humans or animal experiments.

Response: Thanks for the comments. Figure 2 was created by the program String, which does not allow us make that much change. But we added more description in the figure legend . We added more comments in the text on the experiments.

Reviewer 2 Report

In this study demonstrated the role of on target genes for Cholesterol Regulation in the brain to contest Alzheimer ’s disease. The results of this study selected GSK3B, BACE1, SREBP2, CYP46A1, ABCA1, and TREM2 were the top target genes. These proteins synthesized/transported cholesterol and produced amyloid-beta  and/or tau proteins. SREBP2, BACE1, and GSK3B were chosen for inhibition, while ABCA1, CYP46A1, and TREM2 were chosen for activation. It is interestingly informative in its present version and can add some novelty to the community dealing with the Alzheimer’s disease investigation and diagnosis. I recommend publication in its present form.

Author Response

Response: Thanks for the positive comments.

Reviewer 3 Report

The manuscript entitled “Investigation of Potential Drug Targets for Cholesterol Regulation to Treat Alzheimer’s Disease” by Marina Passero et al. provides a comprehensive summary of advancements in investigation of potential drug targets for brain cholesterol regulation. The manuscript majorly focuses on protein-protein interaction analysis, pathways associated with brain cholesterol biosynthesis and existing clinical trials data of potential targets. This review is very interesting. The study is comprehensive with solid literature evidence and manuscript is well written.

Minor suggestions:

1.      In abstract and conclusions, authors stated that “SREBP2 was recommended as the top protein target, as it has a well-studied crystal structure that can be fed to existing ligand-protein docking programs for in silico screening of chemical compounds”. Although it is relevant, it would be more compelling to include your stronger literature findings on the rationale for choosing the target in abstract and conclusion to catch the reader’s attention.

2.      In fact, the manuscript lacking the substantial information about crystal structure data. Authors are suggested to include structural aspects for the mentioned targets.

3.      The discussion part is well written. However, it would be helpful to the readers if the authors include viewpoint of selecting SREBP2 target, instead of focusing on other targets.

In summary, I recommend this manuscript for acceptance in International Journal of Environmental Research and Public Health after some minor alteration.

Author Response

The manuscript entitled “Investigation of Potential Drug Targets for Cholesterol Regulation to Treat Alzheimer’s Disease” by Marina Passero et al. provides a comprehensive summary of advancements in investigation of potential drug targets for brain cholesterol regulation. The manuscript majorly focuses on protein-protein interaction analysis, pathways associated with brain cholesterol biosynthesis and existing clinical trials data of potential targets. This review is very interesting. The study is comprehensive with solid literature evidence and manuscript is well written.

Minor suggestions:

  1. In abstract and conclusions, authors stated that “SREBP2 was recommended as the top protein target, as it has a well-studied crystal structure that can be fed to existing ligand-protein docking programs for in silico screening of chemical compounds”. Although it is relevant, it would be more compelling to include your stronger literature findings on the rationale for choosing the target in abstract and conclusion to catch the reader’s attention.

Response: Thanks for the comments. We have added more comments in the Abstract and Introduction according to your suggestion.

  1. In fact, the manuscript lacking the substantial information about crystal structure data. Authors are suggested to include structural aspects for the mentioned targets.

Response: Thanks for the comments. We have added more information about the crystal structure of SREBP2. 3. One structure (7AZN) listed in SREBP2 was removed, which is Aster proteins (encoded by the Gramd1a-c genes) rather than SREBP2.

  1. The discussion part is well written. However, it would be helpful to the readers if the authors include viewpoint of selecting SREBP2 target, instead of focusing on other targets.

Response: Thanks for the comments. We have revised the manuscript to elaborate on the pros and cons of targeting the other proteins and provide a clear comparison that justifies our selection of SREBP2.

In summary, I recommend this manuscript for acceptance in International Journal of Environmental Research and Public Health after some minor alteration.

Response: Thanks for the positive comments

Reviewer 4 Report

Line 7 in abstract: only seven drugs –which seven drugs-need to be clarified for general readers or change to several drugs

Figure 1 is very obscure to understand

Author Response

Line 7 in abstract: only seven drugs –which seven drugs-need to be clarified for general readers or change to several drugs

Response: we have rewritten the first two sentence to avoid confusion: Despite extensive research and seven approved drugs, the complex interplay of genes, proteins, and pathways in Alzheimer’s disease remains a challenge. The seven drugs were discussed in detail in the main text.

Figure 1 is very obscure to understand

Response: We have added labels onto the lines indicating the nature of the interaction (e.g., activation, repression) in Figure 2.

We also cited more references in the manuscript.